# Semigroup Structures and Commutative Ideals of BCK-Algebras Based on Crossing Cubic Set Structures

**Mehmet Ali Öztürk** [1,*], **Damla Yılmaz** [2] **and Young Bae Jun** [3]

1   Department of Mathematics, Faculty of Arts and Sciences, Adıyaman University, Adıyaman 02040, Turkey;
2   Department of Mathematics, Faculty of Sciences, Erzurum Technical University, Erzurum 25050, Turkey; damla.yilmaz@erzurum.edu.tr
3   Department of Mathematics Education, Gyeongsang National University, Jinju 52828, Korea; skywine@gmail.com
*   Correspondence: mehaliozturk@gmail.com

**Abstract:** First, semigroup structure is constructed by providing binary operations for the crossing cubic set structure. The concept of commutative crossing cubic ideal is introduced by applying crossing cubic set structure to commutative ideal in BCK-algebra, and several properties are investigated. The relationship between crossing cubic ideal and commutative crossing cubic ideal is discussed. An example to show that crossing cubic ideal is not commutative crossing cubic ideal is given, and then the conditions in which crossing cubic ideal can be commutative crossing cubic ideal are explored. Characterizations of commutative crossing cubic ideal are discussed, and the relationship between commutative crossing cubic ideal and crossing cubic level set is considered. An extension property of commutative crossing cubic ideal is established, and the translation of commutative crossing cubic ideal is studied. Conditions for the translation of crossing cubic set structure to be commutative crossing cubic ideal are provided, and its characterization is processed.

**Keywords:** crossing cubic subalgebra; crossing cubic ideal; commutative crossing cubic ideal; translation

## 1. Introduction

A crisp set $A$ in a universe $K$ can be described as its characteristic function $\xi_A : K \rightarrow \{0, 1\}$. If we take an extension $[0, 1]$ of the range $\{0, 1\}$ in the characteristic function $\xi_A : K \rightarrow \{0, 1\}$, we can obtain a new function $\xi : K \rightarrow [0, 1]$, and it is called the fuzzy set, which is introduced by Zadeh [1]. A fuzzy set is a very useful and valuable tool for processing positive information while having limitations in processing negative information. Based on the need for tools necessary to process negative information, Jun et al. [2] introduced the negative-valued function and applied it to the BCK/BCI-algebra. They discussed the ideal theory of BCK/BCI-algebras based on negative-valued functions. The interval-valued fuzzy set, which is defined by an interval-valued membership function, was proposed more than 40 years ago as a natural extension of the set of fuzzy. The fuzzy set theory and interval-valued fuzzy set theory are good mathematical tools for dealing with uncertainty in a parametric manner and are widely applied to pure mathematics, medical diagnosis, decision-making, etc. (see [3–8]). Yang et al. [9] attempted to combine the interval value fuzzy set and [10] the soft set for the purpose of obtaining a new soft set model, and then they used it to analyze a decision-making problem. The Pythagorean fuzzy set and interval value Pythagorean fuzzy set play an important role in the decision-making process. Abbas et al. [11] introduced the concept of a cubic Pythagorean fuzzy set based on the Pythagorean fuzzy set and the interval value Pythagorean fuzzy set. They developed a cubic Pythagorean fuzzy weighted mean operator and a cubic Pythagorean fuzzy weighted geometric operator, and they applied it to multi-attribute decision-making with unknown

weight information. The spherical fuzzy sets and interval-valued spherical fuzzy sets are better methods to deal with fuzziness and uncertainty. Based on the spherical fuzzy sets and interval-valued spherical fuzzy sets, Ayaz et al. [12] introduced the spherical cubic fuzzy set and discussed various aggregation operators. Moreover, its application in multi-attribute decision-making is also discussed by Tehreem et al. [13]. In the social structure, where the development of science and the occurrence of complex problems involving various factors are frequent, we feel the need for new tools in the expectation that one tool will have limitations in solving these problems. Moreover, it can be observed that the cubic set theory is useful in dealing with the problem of uncertainty about positive information, but there is a limit to dealing with the problem of uncertainty, including negative information. Therefore, Jun et al. [14] introduced crossing cubic set structure as an extension of bipolar fuzzy set, consisting of interval-valued fuzzy set and negative-valued function, and applied it to BCK-algebra. In [15], Jun et al. introduced the crossing cubic ideal in BCK/BCI-algebra, closed crossing cubic ideal in BCI-algebra, and crossing cubic ∘-subalgebra of BCK-algebra with the condition (S). They identified the relationship between them. They provided conditions for crossing cubic structure to be closed crossing cubic ideal, and explored the conditions under which crossing cubic ideal is closed. They discussed characterizations of crossing cubic ideal, and we studied the translation of crossing cubic subalgebra and crossing cubic ideal.

In this paper, we first construct a semigroup structure by giving binary operations on the set of crossing cubic set structures. We apply the crossing cubic set structure to commutative ideals of BCK-algebras, and define commutative crossing cubic ideal. We investigate the relationship between crossing cubic ideal and commutative crossing cubic ideal. We first present an example in which crossing cubic ideal is not commutative crossing cubic ideal, and explore the conditions in which crossing cubic ideal can be commutative crossing cubic ideal. We discuss characterizations of commutative crossing cubic ideal, and investigate the relationship between commutative crossing cubic ideal and crossing cubic level set. We establish an extension property of commutative crossing cubic ideal. We study the translation of commutative crossing cubic ideal. We find conditions for the translation of crossing cubic set structure to be commutative crossing cubic ideal, and consider its characterization.

## 2. Preliminaries

A set $K$ with a special element 0 and a binary operation "$\div$" is called *BCI-algebra* if it satisfies the following:

$$(\forall a, b, c \in K)(((a \div b) \div (a \div c)) \div (c \div b) = 0), \tag{1}$$

$$(\forall a, b \in K)((a \div (a \div b)) \div b = 0), \tag{2}$$

$$(\forall a \in K)(a \div a = 0), \tag{3}$$

$$(\forall a, b \in K)(a \div b = 0, \, b \div a = 0 \, \Rightarrow \, a = b). \tag{4}$$

A *BCK-algebra* is defined as a BCI-algebra $K$ satisfying the following condition:

$$(\forall a \in K)(0 \div a = 0). \tag{5}$$

Every BCI/BCK-algebra $K$ satisfies the following:

$$(\forall a \in K)(a \div 0 = a), \tag{6}$$

$$(\forall a, b, c \in K)((a \div b) \div c = (a \div c) \div b), \tag{7}$$

$$(\forall a, b, c \in K)(a \leq b \, \Rightarrow \, a \div c \leq b \div c, \, c \div b \leq c \div a), \tag{8}$$

A BCK-algebra $K$ is said to be *commutative* (see [16]) if $a \wedge b = b \wedge a$ for all $a, b \in K$ where $a \wedge b = a \div (a \div b)$. We will abbreviate commutative BCK-algebra to cBCK-algebra.

A subset $L$ of a BCI/BCK-algebra $K$ is called the following:

- A *subalgebra* of $K$ (see [16,17]) if $a \div b \in L$ for all $a, b \in L$.
- An *ideal* of $K$ (see [16,17]) if it satisfies

$$0 \in L, \tag{9}$$

$$(\forall a, b \in K)(a \div b \in L, \ b \in L \ \Rightarrow \ a \in L). \tag{10}$$

A subset $L$ of a BCK-algebra $K$ is called a *commutative ideal* of $K$ (see [18]) if it satisfies Equation (9) and

$$(\forall a, b, c \in K)((a \div b) \div c \in L, \ c \in L \ \Rightarrow \ a \div (b \wedge a) \in L). \tag{11}$$

We denote by $\mathcal{F}(K, [-1,0])$ the collection of all functions from a set $K$ to $[-1,0]$. We say that an element of $\mathcal{F}(K, [-1,0])$ is a *negative-valued function* from $K$ to $[-1,0]$ (briefly, $\mathcal{N}$-*function* on $K$). We define a relation $\leq$ on $\mathcal{F}(K, [-1,0])$ as follows:

$$\zeta \leq \eta \ \Leftrightarrow \ (\forall a \in K)(\zeta(a) \leq \eta(a)) \tag{12}$$

for all $\zeta, \eta \in \mathcal{F}(K, [-1,0])$.

An *interval number* is defined to be a subinterval $\tilde{a} = [a^-, a^+]$ of $[0,1]$, where $0 \leq a^- \leq a^+ \leq 1$. The interval numbers $[0,0]$ and $[1,1]$ are denoted by $\tilde{0}$ and $\tilde{1}$, respectively. We denote by $[[0,1]]$ the set of all interval numbers. Let us define what is known as *refined minimum* (briefly, rmin) of two elements in $[[0,1]]$. We also define the symbols "$\succeq$", "$\preceq$", "$=$" in case of two elements in $[[0,1]]$. Consider two interval numbers $\tilde{a}_1 := [a_1^-, a_1^+]$ and $\tilde{a}_2 := [a_2^-, a_2^+]$. Then,

$$\mathrm{rmin}\{\tilde{a}_1, \tilde{a}_2\} = \left[\min\{a_1^-, a_2^-\}, \min\{a_1^+, a_2^+\}\right],$$
$$\tilde{a}_1 \succeq \tilde{a}_2 \ \Leftrightarrow \ a_1^- \geq a_2^-, \ a_1^+ \geq a_2^+,$$

and, similarly, we may have $\tilde{a}_1 \preceq \tilde{a}_2$ and $\tilde{a}_1 = \tilde{a}_2$. To say $\tilde{a}_1 \succ \tilde{a}_2$ (resp. $\tilde{a}_1 \prec \tilde{a}_2$), we mean $\tilde{a}_1 \succeq \tilde{a}_2$ and $\tilde{a}_1 \neq \tilde{a}_2$ (resp. $\tilde{a}_1 \preceq \tilde{a}_2$ and $\tilde{a}_1 \neq \tilde{a}_2$). Let $\tilde{a}_i \in [[0,1]]$ where $i \in \Lambda$. We define

$$\operatorname*{rinf}_{i \in \Lambda} \tilde{a}_i = \left[\inf_{i \in \Lambda} a_i^-, \inf_{i \in \Lambda} a_i^+\right] \quad \text{and} \quad \operatorname*{rsup}_{i \in \Lambda} \tilde{a}_i = \left[\sup_{i \in \Lambda} a_i^-, \sup_{i \in \Lambda} a_i^+\right].$$

Let $K$ be a nonempty set. A function $\hbar : K \to [[0,1]]$ is called an *interval-valued fuzzy set* (briefly, an *IVF set*) in $K$. Let $[[0,1]]^K$ stand for the set of all IVF sets in $K$. Every $\hbar \in [[0,1]]^K$ and $a \in K$, $\hbar(a) = [\hbar^-(a), \hbar^+(a)]$ is called the *degree* of membership of an element $a$ to $\hbar$, where $\hbar^- : K \to [0,1]$ and $\hbar^+ : K \to [0,1]$ are fuzzy sets in $K$ which are called a *lower fuzzy set* and an *upper fuzzy set* in $K$, respectively. For simplicity, we denote $\hbar = [\hbar^-, \hbar^+]$. For every $\hbar, \pounds \in [[0,1]]^K$, we define

$$\hbar \subseteq \pounds \ \Leftrightarrow \ \hbar(a) \preceq \pounds(a) \ \text{for all } a \in K,$$

and

$$\hbar = \pounds \ \Leftrightarrow \ \hbar(a) = \pounds(a) \ \text{for all } a \in K.$$

**Definition 1** ([14])**.** *By a crossing cubic set structure on a set $K$, we mean a pair $\left(K, \mathcal{C}_{(\hbar, \zeta)}\right)$ where*

$$\mathcal{C}_{(\hbar, \zeta)} := \{\langle a, \hbar(a), \zeta(a) \rangle \mid a \in K\} \tag{13}$$

*in which $\hbar$ is an interval-valued fuzzy set in $K$ and $\zeta$ is an $\mathcal{N}$-function on $K$.*

**Definition 2** ([14])**.** *A crossing cubic set structure* $\left( K, \mathcal{C}_{(\hbar,\zeta)} \right)$ *on a BCI/BCK-algebra K is called a crossing cubic subalgebra of K if it satisfies the following:*

$$(\forall a, b \in K) \left( \begin{array}{l} \hbar(a \div b) \succeq \mathrm{rmin}\{\hbar(a), \hbar(b)\} \\ \zeta(a \div b) \leq \max\{\zeta(a), \zeta(b)\} \end{array} \right). \tag{14}$$

**Proposition 1** ([14])**.** *Every crossing cubic subalgebra* $\left( K, \mathcal{C}_{(\hbar,\zeta)} \right)$ *of a BCI/BCK-algebra K satisfies*

$$(\forall a \in K)(\hbar(0) \succeq \hbar(a), \zeta(0) \leq \zeta(a)). \tag{15}$$

**Definition 3** ([15])**.** *A crossing cubic set structure* $\left( K, \mathcal{C}_{(\hbar,\zeta)} \right)$ *on a BCI/BCK-algebra K is called a crossing cubic ideal (briefly, CC-ideal) of K if it satisfies the following:*

$$(\forall x, y \in K) \left( \begin{array}{l} \hbar(0) \succeq \hbar(x) \succeq \mathrm{rmin}\{\hbar(x \div y), \hbar(y)\} \\ \zeta(0) \leq \zeta(x) \leq \max\{\zeta(x \div y), \zeta(y)\} \end{array} \right). \tag{16}$$

## 3. Semigroup Structures of Crossing Cubic Set Structures

We denote by $CCS(K)$ the collection of crossing cubic set structures on a set $K$.

**Definition 4.** *Let* $\left( K, \mathcal{C}_{(\hbar,\zeta)} \right), \left( K, \mathcal{C}_{(\pounds,\eta)} \right) \in CCS(K)$*. Then, the sum of* $\left( K, \mathcal{C}_{(\hbar,\zeta)} \right)$ *and* $\left( K, \mathcal{C}_{(\pounds,\eta)} \right)$*, denoted by* $\left( K, \mathcal{C}_{(\hbar,\zeta)} \right) \oplus \left( K, \mathcal{C}_{(\pounds,\eta)} \right)$*, is defined as a crossing cubic set structure* $\left( K, \mathcal{C}_{(\hbar \oplus \pounds, \zeta \oplus \eta)} \right)$ *in which* $\hbar \oplus \pounds : K \to [[0,1]]$ *and* $\zeta \oplus \eta : K \to [-1,0]$ *are given as follows:*

$$(\hbar \oplus \pounds)(x) = [(\hbar \oplus \pounds)^-(x), (\hbar \oplus \pounds)^+(x)]$$

*and* $(\zeta \oplus \eta)(x) = \zeta(x) + \eta(x) - \zeta(x) \cdot \eta(x)$*, respectively, where*

$$(\hbar \oplus \pounds)^-(x) = \hbar^-(x) + \pounds^-(x) - \hbar^-(x) \cdot \pounds^-(x)$$

*and* $(\hbar \oplus \pounds)^+(x) = \hbar^+(x) + \pounds^+(x) - \hbar^+(x) \cdot \pounds^+(x)$*.*

**Example 1.** *Let* $K = [0,1]$ *and we define*

$$\hbar : K \to [[0,1]], \ x \mapsto [0.9x, x],$$
$$\zeta : K \to [-1,0], \ x \mapsto -x$$

*and*

$$\pounds : K \to [[0,1]], \ x \mapsto [0.5x^2, x^2],$$
$$\eta : K \to [-1,0], \ x \mapsto -\tfrac{1}{2}x.$$

*Then,* $\left( K, \mathcal{C}_{(\hbar,\zeta)} \right), \left( K, \mathcal{C}_{(\pounds,\eta)} \right) \in CCS(K)$*, and their sum*

$$\left( K, \mathcal{C}_{(\hbar,\zeta)} \right) \oplus \left( K, \mathcal{C}_{(\pounds,\eta)} \right) = \left( K, \mathcal{C}_{(\hbar \oplus \pounds, \zeta \oplus \eta)} \right)$$

*is given as follows:*

$$\hbar \oplus \pounds : K \to [[0,1]], \ x \mapsto [0.9x + 0.5x^2 - 0.45x^3, x + x^2 - x^3],$$
$$\zeta \oplus \eta : K \to [-1,0], \ x \mapsto -x - \tfrac{1}{2}x - \tfrac{1}{2}x^2.$$

Let $\left(K, \mathcal{C}_{(\hbar,\zeta)}\right), \left(K, \mathcal{C}_{(\pounds,\eta)}\right), \left(K, \mathcal{C}_{(\eth,\xi)}\right) \in CCS(K)$. Then,

$$(\hbar \oplus \pounds)^-(x) + \eth^-(x) - (\hbar \oplus \pounds)^-(x) \cdot \eth^-(x)$$
$$= \hbar^-(x) + \pounds^-(x) - \hbar^-(x) \cdot \pounds^-(x) + \eth^-(x)$$
$$\quad - (\hbar^-(x) + \pounds^-(x) - \hbar^-(x) \cdot \pounds^-(x)) \cdot \eth^-(x)$$
$$= \hbar^-(x) + \pounds^-(x) - \hbar^-(x) \cdot \pounds^-(x) + \eth^-(x)$$
$$\quad - \hbar^-(x) \cdot \eth^-(x) - \pounds^-(x) \cdot \eth^-(x) + \hbar^-(x) \cdot \pounds^-(x) \cdot \eth^-(x),$$

$$\hbar^-(x) + (\pounds^- \oplus \eth^-)(x) - \hbar^-(x) \cdot (\pounds^- \oplus \eth^-)(x)$$
$$= \hbar^-(x) + (\pounds^-(x) + \eth^-(x) - \pounds^-(x) \cdot \eth^-(x))$$
$$\quad - \hbar^-(x) \cdot (\pounds^-(x) + \eth^-(x) - \pounds^-(x) \cdot \eth^-(x))$$
$$= \hbar^-(x) + \pounds^-(x) + \eth^-(x) - \pounds^-(x) \cdot \eth^-(x)$$
$$\quad - \hbar^-(x) \cdot \pounds^-(x) - \hbar^-(x) \cdot \eth^-(x) + \hbar^-(x) \cdot \pounds^-(x) \cdot \eth^-(x),$$

$$(\hbar \oplus \pounds)^+(x) + \eth^+(x) - (\hbar \oplus \pounds)^+(x) \cdot \eth^+(x)$$
$$= \hbar^+(x) + \pounds^+(x) - \hbar^+(x) \cdot \pounds^+(x) + \eth^+(x)$$
$$\quad - (\hbar^+(x) + \pounds^+(x) - \hbar^+(x) \cdot \pounds^+(x)) \cdot \eth^+(x)$$
$$= \hbar^+(x) + \pounds^+(x) - \hbar^+(x) \cdot \pounds^+(x) + \eth^+(x)$$
$$\quad - \hbar^+(x) \cdot \eth^+(x) - \pounds^+(x) \cdot \eth^+(x) + \hbar^+(x) \cdot \pounds^+(x) \cdot \eth^+(x),$$

and

$$\hbar^+(x) + (\pounds^+ \oplus \eth^+)(x) - \hbar^+(x) \cdot (\pounds^+ \oplus \eth^+)(x)$$
$$= \hbar^+(x) + (\pounds^+(x) + \eth^+(x) - \pounds^+(x) \cdot \eth^+(x))$$
$$\quad - \hbar^+(x) \cdot (\pounds^+(x) + \eth^+(x) - \pounds^+(x) \cdot \eth^+(x))$$
$$= \hbar^+(x) + \pounds^+(x) + \eth^+(x) - \pounds^+(x) \cdot \eth^+(x)$$
$$\quad - \hbar^+(x) \cdot \pounds^+(x) - \hbar^+(x) \cdot \eth^+(x) + \hbar^+(x) \cdot \pounds^+(x) \cdot \eth^+(x),$$

that is, $(\hbar \oplus \pounds)^-(x) + \eth^-(x) - (\hbar \oplus \pounds)^-(x) \cdot \eth^-(x) = \hbar^-(x) + (\pounds^- \oplus \eth^-)(x) - \hbar^-(x) \cdot (\pounds^- \oplus \eth^-)(x)$ and $(\hbar \oplus \pounds)^+(x) + \eth^+(x) - (\hbar \oplus \pounds)^+(x) \cdot \eth^+(x) = \hbar^+(x) + (\pounds^+ \oplus \eth^+)(x) - \hbar^+(x) \cdot (\pounds^+ \oplus \eth^+)(x)$ for all $x \in K$. Hence, $(\hbar \oplus \pounds) \oplus \eta = \hbar \oplus (\pounds \oplus \eta)$. In addition, we obtain

$$((\zeta \oplus \eta) \oplus \xi)(x) = (\zeta \oplus \eta)(x) + \xi(x) - (\zeta \oplus \eta)(x) \cdot \xi(x)$$
$$= \zeta(x) + \eta(x) - \zeta(x) \cdot \eta(x) + \xi(x) - (\zeta(x) + \eta(x) - \zeta(x) \cdot \eta(x)) \cdot \xi(x)$$
$$= \zeta(x) + \eta(x) - \zeta(x) \cdot \eta(x) + \xi(x) - \zeta(x) \cdot \xi(x) - \eta(x) \cdot \xi(x) + \zeta(x) \cdot \eta(x) \cdot \xi(x)$$
$$= \zeta(x) + \eta(x) + \xi(x) - \eta(x) \cdot \xi(x) - \zeta(x) \cdot \eta(x) - \zeta(x) \cdot \xi(x) + \zeta(x) \cdot \eta(x) \cdot \xi(x)$$
$$= \zeta(x) + (\eta(x) + \xi(x) - \eta(x) \cdot \xi(x)) - \zeta(x) \cdot (\eta(x) + \xi(x) - \eta(x) \cdot \xi(x))$$
$$= \zeta(x) + (\eta \oplus \xi)(x) - \zeta(x) \cdot (\eta \oplus \xi)(x)$$
$$= (\zeta \oplus (\eta \oplus \xi))(x)$$

for all $x \in K$. Thus,

$$\left(\left(K, \mathcal{C}_{(\hbar,\zeta)}\right) \oplus \left(K, \mathcal{C}_{(\pounds,\eta)}\right)\right) \oplus \left(K, \mathcal{C}_{(\eth,\xi)}\right) = \left(K, \mathcal{C}_{(\hbar,\zeta)}\right) \oplus \left(\left(K, \mathcal{C}_{(\pounds,\eta)}\right) \oplus \left(K, \mathcal{C}_{(\eth,\xi)}\right)\right).$$

Now, we have

$$(\hbar \oplus £)^-(x) = \hbar^-(x) + £^-(x) - \hbar^-(x) \cdot £^-(x)$$
$$= £^-(x) + \hbar^-(x) - £^-(x) \cdot \hbar^-(x)$$
$$= (£ \oplus \hbar)^-(x),$$

$$(\hbar \oplus £)^+(x) = \hbar^+(x) + £^+(x) - \hbar^+(x) \cdot £^+(x)$$
$$= £^+(x) + \hbar^+(x) - £^+(x) \cdot \hbar^+(x)$$
$$= (£ \oplus \hbar)^+(x),$$

and $(\zeta \oplus \eta)^-(x) = \zeta^-(x) + \eta^-(x) - \zeta^-(x) \cdot \eta^-(x) = \eta^-(x) + \zeta^-(x) - \eta^-(x) \cdot \zeta^-(x) = (\eta \oplus \zeta)^-(x)$ for all $x \in K$. Hence, $\left(K, \mathcal{C}_{(\hbar,\zeta)}\right) \oplus \left(K, \mathcal{C}_{(£,\eta)}\right) = \left(K, \mathcal{C}_{(£,\eta)}\right) \oplus \left(K, \mathcal{C}_{(\hbar,\zeta)}\right)$.

Consider a crossing cubic set structure $\left(K, \mathcal{C}_{(\tilde{0},\mathbf{0})}\right)$ on $K$ in which

$$\tilde{0} : K \to [[0,1]], \ x \mapsto [0,0]$$

and

$$\mathbf{0} : K \to [-1,0], \ x \mapsto 0.$$

For every $\left(K, \mathcal{C}_{(\hbar,\zeta)}\right) \in CCS(K)$, we obtain

$$(\hbar \oplus \tilde{0})(x) = [\hbar^-(x) + \tilde{0}^-(x) - \hbar^-(x) \cdot \tilde{0}^-(x), \ \hbar^+(x) + \tilde{0}^+(x) - \hbar^+(x) \cdot \tilde{0}^+(x)]$$
$$= [\hbar^-(x), \hbar^+(x)],$$

and $(\zeta \oplus \mathbf{0})(x) = \zeta(x) + \mathbf{0}(x) - \zeta(x) \cdot \mathbf{0}(x) = \zeta(x)$ for all $x \in K$. Similarly, $(\tilde{0} \oplus \hbar)(x) = [\hbar^-(x), \hbar^+(x)]$ and $(\mathbf{0} \oplus \zeta)(x) = \zeta(x)$ for all $x \in K$. Thus,

$$\left(K, \mathcal{C}_{(\hbar,\zeta)}\right) \oplus \left(K, \mathcal{C}_{(\tilde{0},\mathbf{0})}\right) = \left(K, \mathcal{C}_{(\hbar,\zeta)}\right) = \left(K, \mathcal{C}_{(\tilde{0},\mathbf{0})}\right) \oplus \left(K, \mathcal{C}_{(\hbar,\zeta)}\right).$$

Based on the above calculations, we obtain the next theorem.

**Theorem 1.** *The collection $CCS(K)$ is a commutative monoid under the operation " $\oplus$ " with identity $\left(K, \mathcal{C}_{(\tilde{0},\mathbf{0})}\right)$.*

**Definition 5.** *Let $\left(K, \mathcal{C}_{(\hbar,\zeta)}\right), \left(K, \mathcal{C}_{(£,\eta)}\right) \in CCS(K)$. Then, the product of $\left(K, \mathcal{C}_{(\hbar,\zeta)}\right)$ and $\left(K, \mathcal{C}_{(£,\eta)}\right)$, denoted by $\left(K, \mathcal{C}_{(\hbar,\zeta)}\right) \otimes \left(K, \mathcal{C}_{(£,\eta)}\right)$, is defined as a crossing cubic set structure $\left(K, \mathcal{C}_{(\hbar \otimes £, \zeta \otimes \eta)}\right)$ where*

$$\hbar \otimes £ : K \to [[0,1]], \ x \mapsto [\hbar^-(x) \cdot £^-(x), \ \hbar^+(x) \cdot £^+(x)],$$
$$\zeta \otimes \eta : K \to [-1,0], \ x \mapsto -\zeta(x) \cdot \eta(x).$$

**Example 2.** *Consider the crossing cubic set structures $\left(K, \mathcal{C}_{(\hbar,\zeta)}\right)$ and $\left(K, \mathcal{C}_{(£,\eta)}\right)$ in Example 1. Then, their product*

$$\left(K, \mathcal{C}_{(\hbar,\zeta)}\right) \otimes \left(K, \mathcal{C}_{(£,\eta)}\right) = \left(K, \mathcal{C}_{(\hbar \otimes £, \zeta \otimes \eta)}\right)$$

*is given as follows:*

$$\hbar \otimes \text{£} : K \to [[0,1]], \ x \mapsto [0.45x^3, x^3],$$

$$\zeta \otimes \eta : K \to [-1,0], \ x \mapsto -\tfrac{1}{2}x^2.$$

**Theorem 2.** *The collection $CCS(K)$ is a commutative semigroup under the operation "$\otimes$".*

**Proof.** Let $\left(K, \mathcal{C}_{(\hbar,\zeta)}\right), \left(K, \mathcal{C}_{(\text{£},\eta)}\right), \left(K, \mathcal{C}_{(\eth,\xi)}\right) \in CCS(K)$. Then,

$$
\begin{aligned}
((\hbar \otimes \text{£}) \otimes \eth)(x) &= [(\hbar \otimes \text{£})^-(x) \cdot \eth^-(x), (\hbar \otimes \text{£})^+(x) \cdot \eth^+(x)] \\
&= [(\hbar^-(x) \cdot \text{£}^-(x)) \cdot \eth^-(x), (\hbar^+(x) \cdot \text{£}^+(x)) \cdot \eth^+(x)] \\
&= [\hbar^-(x) \cdot (\text{£}^-(x) \cdot \eth^-(x)), \hbar^+(x) \cdot (\text{£}^+(x) \cdot \eth^+(x))] \\
&= [\hbar^-(x) \cdot (\text{£} \otimes \eth)^-(x), \hbar^+(x) \cdot (\text{£} \otimes \eth)^+(x)] \\
&= (\hbar \otimes (\text{£} \otimes \eth))(x)
\end{aligned}
$$

*and*

$$
\begin{aligned}
((\zeta \otimes \eta) \otimes \xi)(x) &= (\zeta \otimes \eta)(x) \cdot \xi(x) = (\zeta(x) \cdot \eta(x)) \cdot \xi(x) \\
&= \zeta(x) \cdot (\eta(x) \cdot \xi(x)) = \zeta(x) \cdot (\eta \otimes \xi)(x) \\
&= (\zeta \otimes (\eta \otimes \xi))(x)
\end{aligned}
$$

*for all $x \in K$. Hence,*

$$\left(\left(K, \mathcal{C}_{(\hbar,\zeta)}\right) \otimes \left(K, \mathcal{C}_{(\text{£},\eta)}\right)\right) \otimes \left(K, \mathcal{C}_{(\eth,\xi)}\right) = \left(K, \mathcal{C}_{(\hbar,\zeta)}\right) \otimes \left(\left(K, \mathcal{C}_{(\text{£},\eta)}\right) \otimes \left(K, \mathcal{C}_{(\eth,\xi)}\right)\right).$$

In addition, we have

$$(\hbar \otimes \text{£})(x) = [\hbar^-(x) \cdot \text{£}^-(x), \hbar^+(x) \cdot \text{£}^+(x)] = [\text{£}^-(x) \cdot \hbar^-(x), \text{£}^+(x) \cdot \hbar^+(x)] = (\text{£} \otimes \hbar)(x)$$

and

$$(\zeta \otimes \eta)(x) = [\zeta^-(x) \cdot \eta^-(x), \zeta^+(x) \cdot \eta^+(x)] = [\eta^-(x) \cdot \zeta^-(x), \eta^+(x) \cdot \zeta^+(x)] = (\eta \otimes \zeta)(x)$$

for all $x \in K$. Thus, $\left(K, \mathcal{C}_{(\hbar,\zeta)}\right) \otimes \left(K, \mathcal{C}_{(\text{£},\eta)}\right) = \left(K, \mathcal{C}_{(\text{£},\eta)}\right) \otimes \left(K, \mathcal{C}_{(\hbar,\zeta)}\right)$ and therefore $(CCS(K), \otimes)$ is a commutative semigroup. □

## 4. Crossing Cubic Set Structure Applied to Commutative Ideals in BCK-Algebras

In the following, let $K$ denote a BCK-algebra unless otherwise specified.

**Definition 6.** *A crossing cubic set structure $\left(K, \mathcal{C}_{(\hbar,\zeta)}\right)$ on K is called a commutative crossing cubic ideal (briefly, cCC-ideal) of K if it satisfies*

$$(\forall x \in K)\left( \ \hbar(0) \succeq \hbar(x), \ \zeta(0) \le \zeta(x) \ \right) \tag{17}$$

*and*

$$(\forall x,y,z \in K)\left( \begin{array}{l} \hbar(x \div (y \wedge x)) \succeq \text{rmin}\{\hbar((x \div y) \div z), \hbar(z)\} \\ \zeta(x \div (y \wedge x)) \le \max\{\zeta((x \div y) \div z), \zeta(z)\} \end{array} \right). \tag{18}$$

If *K* is commutative, then the condition Equation (18) is equivalent to the next condition:

$$(\forall x, y, z \in K) \left( \begin{array}{l} \hbar(x \div (x \wedge y)) \succeq \text{rmin}\{\hbar((x \div y) \div z), \hbar(z)\} \\ \zeta(x \div (x \wedge y)) \leq \max\{\zeta((x \div y) \div z), \zeta(z)\} \end{array} \right). \qquad (19)$$

**Example 3.** *Let* $K = \{0, \varrho_1, \varrho_2, \varrho_3\}$ *be a set with the binary operation "$\div$" which is given in Table 1.*

**Table 1.** Cayley table for the binary operation "$\div$".

| $\div$ | 0 | $\varrho_1$ | $\varrho_2$ | $\varrho_3$ |
|---|---|---|---|---|
| 0 | 0 | 0 | 0 | 0 |
| $\varrho_1$ | $\varrho_1$ | 0 | 0 | $\varrho_1$ |
| $\varrho_2$ | $\varrho_2$ | $\varrho_1$ | 0 | $\varrho_2$ |
| $\varrho_3$ | $\varrho_3$ | $\varrho_3$ | $\varrho_3$ | 0 |

Then, *K* is a BCK-algebra (see [16]). Let $\left(K, \mathcal{C}_{(\hbar, \zeta)}\right)$ be a crossing cubic set structure on K, which is given by Table 2.

**Table 2.** Tabular representation for $\left(K, \mathcal{C}_{(\hbar, \zeta)}\right)$.

| *K* | $\hbar(x)$ | $\zeta(x)$ |
|---|---|---|
| 0 | $[0.39, 0.76]$ | $-0.8$ |
| $\varrho_1$ | $[0.23, 0.63]$ | $-0.4$ |
| $\varrho_2$ | $[0.23, 0.63]$ | $-0.4$ |
| $\varrho_3$ | $[0.15, 0.47]$ | $-0.6$ |

*It is routine to verify that* $\left(K, \mathcal{C}_{(\hbar, \zeta)}\right)$ *is a cCC-ideal of K.*

We discuss the relationship between CC-ideal and cCC-ideal.

**Theorem 3.** *Every cCC-ideal is a CC-ideal.*

**Proof.** Let $\left(K, \mathcal{C}_{(\hbar, \zeta)}\right)$ be a cCC-ideal of *K*. If we place $y = 0$ in Equation (18) and use Equations (5) and (6), then

$$\hbar(x) = \hbar(x \div (0 \wedge x)) \succeq \text{rmin}\{\hbar((x \div 0) \div z), \hbar(z)\} = \text{rmin}\{\hbar(x \div z), \hbar(z)\}$$

and $\zeta(x) = \zeta(x \div (0 \wedge x)) \leq \max\{\zeta((x \div 0) \div z), \zeta(z)\} = \max\{\zeta(x \div z), \zeta(z)\}$ for all $x, z \in K$. Therefore, $\left(K, \mathcal{C}_{(\hbar, \zeta)}\right)$ is a CC-ideal of *K*. $\square$

The example below informs the existence of the CC-ideal, not the cCC-ideal.

**Example 4.** *Let* $K = \{0, \varrho_1, \varrho_2, \varrho_3, \varrho_4\}$ *be a set with the binary operation "$\div$", which is given in Table 3.*

**Table 3.** Cayley table for the binary operation "$\div$".

| $\div$ | 0 | $\varrho_1$ | $\varrho_2$ | $\varrho_3$ | $\varrho_4$ |
|---|---|---|---|---|---|
| 0 | 0 | 0 | 0 | 0 | 0 |
| $\varrho_1$ | $\varrho_1$ | 0 | $\varrho_1$ | 0 | 0 |
| $\varrho_2$ | $\varrho_2$ | $\varrho_2$ | 0 | 0 | 0 |
| $\varrho_3$ | $\varrho_3$ | $\varrho_3$ | $\varrho_3$ | 0 | 0 |
| $\varrho_4$ | $\varrho_4$ | $\varrho_4$ | $\varrho_4$ | $\varrho_3$ | 0 |

*Then, K is a BCK-algebra (see [16]). Let $\left(K, \mathcal{C}_{(\hbar,\zeta)}\right)$ be a crossing cubic set structure on K, which is given by Table 4.*

**Table 4.** Tabular representation for $\left(K, \mathcal{C}_{(\hbar,\zeta)}\right)$.

| K | $\hbar(x)$ | $\zeta(x)$ |
|---|---|---|
| 0 | $[0.42, 0.84]$ | $-0.87$ |
| $\varrho_1$ | $[0.37, 0.74]$ | $-0.74$ |
| $\varrho_2$ | $[0.28, 0.66]$ | $-0.58$ |
| $\varrho_3$ | $[0.17, 0.52]$ | $-0.43$ |
| $\varrho_4$ | $[0.17, 0.52]$ | $-0.43$ |

*It is routine to verify that $\left(K, \mathcal{C}_{(\hbar,\zeta)}\right)$ is a CC-ideal of K, but it is not a cCC-ideal of K because $\hbar(\varrho_2 \div (\varrho_3 \wedge \varrho_2)) = \hbar(\varrho_2) = [0.28, 0.66] \not\succeq [0.42, 0.84] = \text{rmin}\{\hbar((\varrho_2 \div \varrho_3) \div 0), \hbar(0)\}$ and/or $\zeta(\varrho_2 \div (\varrho_3 \wedge \varrho_2)) = \zeta(\varrho_2) = -0.58 \not\leq -0.87 = \max\{\zeta((\varrho_2 \div \varrho_3) \div 0), \zeta(0)\}$.*

We explore the conditions under which a CC-ideal can be a cCC-ideal.

**Lemma 1** ([15]). *Every CC-ideal $\left(K, \mathcal{C}_{(\hbar,\zeta)}\right)$ of K satisfies the following:*

$$(\forall x, y, z \in K)\left( x \div y \leq z \;\Rightarrow\; \left\{ \begin{array}{l} \hbar(x) \succeq \text{rmin}\{\hbar(y), \hbar(z)\} \\ \zeta(x) \leq \max\{\zeta(y), \zeta(z)\} \end{array} \right. \right). \tag{20}$$

**Theorem 4.** *In a cBCK-algebra, every CC-ideal is a cCC-ideal.*

**Proof.** Let $\left(K, \mathcal{C}_{(\hbar,\zeta)}\right)$ be a CC-ideal of a cBCK-algebra K. The combination of Equations (1), (3), (7), and the commutativity of K derive the following:

$$((x \div (y \wedge x)) \div ((x \div y) \div z)) \div z = ((x \div (y \wedge x)) \div z) \div ((x \div y) \div z)$$
$$\leq (x \div (y \wedge x)) \div (x \div y) = (x \wedge y) \div (y \wedge x) = 0,$$

that is, $(x \div (y \wedge x)) \div ((x \div y) \div z) \leq z$ for all $x, y, z \in K$. It follows from Lemma 1 that $\hbar(x \div (y \wedge x)) \succeq \text{rmin}\{\hbar((x \div y) \div z), \hbar(z)\}$ and

$$\zeta(x \div (y \wedge x)) \leq \max\{\zeta((x \div y) \div z), \zeta(z)\}.$$

Therefore, $\left(K, \mathcal{C}_{(\hbar,\zeta)}\right)$ is a cCC-ideal of K. $\square$

**Corollary 1.** *If a BCK-algebra K satisfies any of the following conditions:*

$$(\forall x, y \in K)(x \wedge y \leq y \wedge x), \tag{21}$$
$$(\forall x, y \in K)(x \wedge y = y \div (y \div (x \wedge y))), \tag{22}$$
$$(\forall x, y \in K)(x \leq y \;\Rightarrow\; x = y \wedge x), \tag{23}$$

*then every CC-ideal is a cCC-ideal.*

**Proof.** If a BCK-algebra K satisfies any of three conditions Equations (21)–(23), then K is a cBCK-algebra. Hence every CC-ideal is a cCC-ideal by Theorem 4. $\square$

Note that every BCK-algebra K which is a lower semilattice with respect to the order relation "$\leq$" is a cBCK-algebra. Hence, we have the next corollary.

**Corollary 2.** *If a BCK-algebra K is a lower semilattice with respect to the order relation "$\leq$", then every CC-ideal is a cCC-ideal.*

**Theorem 5.** *Every CC-ideal $\left(K, \mathcal{C}_{(\hbar,\zeta)}\right)$ of K that satisfies*

$$(\forall x, y, z \in K) \left( \begin{array}{c} \hbar((x \div z) \div (y \wedge x)) \succeq \hbar((x \div y) \div z) \\ \zeta((x \div z) \div (y \wedge x)) \leq \zeta((x \div y) \div z) \end{array} \right). \tag{24}$$

*is a cCC-ideal of K.*

**Proof.** Let $\left(K, \mathcal{C}_{(\hbar,\zeta)}\right)$ be a CC-ideal of $K$ that satisfies the condition Equation (24). The combination of Equations (7), (16), and (24) leads to

$$\begin{aligned}
\hbar(x \div (y \wedge x)) &\succeq \mathrm{rmin}\{\hbar((x \div (y \wedge x)) \div z), \hbar(z)\} \\
&= \mathrm{rmin}\{\hbar((x \div z) \div (y \wedge x)), \hbar(z)\} \\
&\succeq \mathrm{rmin}\{\hbar((x \div y) \div z), \hbar(z)\}
\end{aligned}$$

and

$$\begin{aligned}
\zeta(x \div (y \wedge x)) &\leq \max\{\zeta((x \div (y \wedge x)) \div z), \zeta(z)\} \\
&= \max\{\zeta((x \div z) \div (y \wedge x)), \zeta(z)\} \\
&\leq \max\{\zeta((x \div y) \div z), \zeta(z)\}
\end{aligned}$$

for all $x, y, z \in K$. Therefore, $\left(K, \mathcal{C}_{(\hbar,\zeta)}\right)$ is a cCC-ideal of $K$. □

**Theorem 6.** *A crossing cubic set structure $\left(K, \mathcal{C}_{(\hbar,\zeta)}\right)$ on K is a cCC-ideal of K if, and only if, it is a CC-ideal of K that satisfies*

$$(\forall x, y \in K) \left( \begin{array}{c} \hbar(x \div (y \wedge x)) \succeq \hbar(x \div y) \\ \zeta(x \div (y \wedge x)) \leq \zeta(x \div y) \end{array} \right). \tag{25}$$

**Proof.** Assume that $\left(K, \mathcal{C}_{(\hbar,\zeta)}\right)$ is a cCC-ideal of $K$. Then, it is a CC-ideal of $K$ by Theorem 3, and hence we replace $z$ with $0$ in Equation (18) and then use Equations (6) and (16) to find Equation (25).

Conversely, let $\left(K, \mathcal{C}_{(\hbar,\zeta)}\right)$ be a CC-ideal of $K$ that satisfies the condition Equation (25). Then, $\hbar(x \div (y \wedge x)) \succeq \hbar(x \div y) \succeq \mathrm{rmin}\{\hbar((x \div y) \div z), \hbar(z)\}$ and $\zeta(x \div (y \wedge x)) \leq \zeta(x \div y) \leq \max\{\zeta((x \div y) \div z), \zeta(z)\}$ for all $x, y, z \in K$. Therefore, $\left(K, \mathcal{C}_{(\hbar,\zeta)}\right)$ is a cCC-ideal of $K$. □

Given a crossing cubic set structure $\left(K, \mathcal{C}_{(\hbar,\zeta)}\right)$ on $K$ and an element $v \in K$, we consider the next set, which is called a *crossing cubic level set*:

$$K_v := \{x \in K \mid \hbar(x) \succeq \hbar(v), \zeta(x) \leq \zeta(v)\}. \tag{26}$$

**Theorem 7.** *If $\left(K, \mathcal{C}_{(\hbar,\zeta)}\right)$ is a cCC-ideal of K, then the set $K_v$ in Equation (26) is a commutative ideal of K for all $v \in K$.*

**Proof.** Let $v \in K$. It is clear that $0 \in K_v$ by Equation (17). Let $x, y, z \in K$ be such that $z \in K_v$ and $(x \div y) \div z \in K_v$. Then, $\hbar(z) \succeq \hbar(v)$, $\hbar((x \div y) \div z) \succeq \hbar(v)$, $\zeta(z) \leq \zeta(v)$ and $\zeta((x \div y) \div z) \leq \zeta(v)$. It follows from Equation (18) that

$$\hbar(x \div (y \wedge x)) \succeq \mathrm{rmin}\{\hbar((x \div y) \div z), \hbar(z)\} \succeq \hbar(v)$$

and $\zeta(x \div (y \wedge x)) \leq \max\{\zeta((x \div y) \div z), \zeta(z)\} \leq \zeta(v)$. Hence, $x \div (y \wedge x) \in K_v$, and therefore $K_v$ is a commutative ideal of $K$. □

**Proposition 2.** *Given a crossing cubic set structure $\left(K, \mathcal{C}_{(\hbar, \zeta)}\right)$ on $K$, if $K_v$ is a commutative ideal of $K$ for $v \in K$, then the following assertion is valid.*

$$\begin{cases} \mathrm{rmin}\{\hbar((x \div y) \div z), \hbar(z)\} \succeq \hbar(v) \Rightarrow \hbar(x \div (y \wedge x)) \succeq \hbar(v), \\ \max\{\zeta((x \div y) \div z), \zeta(z)\} \leq \zeta(v) \Rightarrow \zeta(x \div (y \wedge x)) \leq \zeta(v). \end{cases} \tag{27}$$

*for all $x, y, z \in K$.*

**Proof.** Assume that $K_v$ is a commutative ideal of $K$ for $v \in K$. Let $x, y, z \in K$ be such that $\mathrm{rmin}\{\hbar((x \div y) \div z), \hbar(z)\} \succeq \hbar(v)$ and $\max\{\zeta((x \div y) \div z), \zeta(z)\} \leq \zeta(v)$. Then, $(x \div y) \div z \in K_v$ and $z \in K_v$, which imply from Equation (11) that $x \div (y \wedge x) \in K_v$. Therefore, $\hbar(x \div (y \wedge x)) \succeq \hbar(v)$ and $\zeta(x \div (y \wedge x)) \leq \zeta(v)$. □

**Theorem 8.** *If a crossing cubic set structure $\left(K, \mathcal{C}_{(\hbar, \zeta)}\right)$ on $K$ satisfies Equations (17) and (27), then the set $K_v$ is a commutative ideal of $K$ for $v \in K$.*

**Proof.** Let $\left(K, \mathcal{C}_{(\hbar, \zeta)}\right)$ be a crossing cubic set structure on $K$ that satisfies Equations (17) and (27). Let $v, x, y, z \in K$ be such that $(x \div y) \div z \in K_v$ and $z \in K_v$. Then, $\hbar((x \div y) \div z) \succeq \hbar(v)$, $\zeta((x \div y) \div z) \leq \zeta(v)$, $\hbar(z) \succeq \hbar(v)$ and $\zeta(z) \leq \zeta(v)$, which imply that $\mathrm{rmin}\{\hbar((x \div y) \div z), \hbar(z)\} \succeq \hbar(v)$ and $\max\{\zeta((x \div y) \div z), \zeta(z)\} \leq \zeta(v)$. Combining these with Equation (27) leads to $\hbar(x \div (y \wedge x)) \succeq \hbar(v)$ and $\zeta(x \div (y \wedge x)) \leq \zeta(v)$. Hence, $x \div (y \wedge x) \in K_v$. The condition Equation (17) induces $0 \in K_v$. Therefore $K_v$ is a commutative ideal of $K$. □

Let $\left(K, \mathcal{C}_{(\hbar, \zeta)}\right)$ be a crossing cubic set structure on $K$. We define a level set of $\left(K, \mathcal{C}_{(\hbar, \zeta)}\right)$, written as $\mathcal{L}\left(K, \mathcal{C}_{(\hbar, \zeta)}\right)$, as follows:

$$\mathcal{L}\left(K, \mathcal{C}_{(\hbar, \zeta)}, \tilde{\beta}, t\right) = \mathcal{L}(K, \hbar, \tilde{\beta}) \cap \mathcal{L}(K, \zeta, t) \tag{28}$$

where $\mathcal{L}(K, \hbar, \tilde{\beta}) = \{x \in K \mid \hbar(x) \succeq \tilde{\beta}\}$ and $\mathcal{L}(K, \zeta, t) = \{x \in K \mid \zeta(x) \leq t\}$ for $\tilde{\beta} := [\beta^-, \beta^+] \in [[0, 1]]$ and $t \in [-1, 0]$. We say that $\mathcal{L}(K, \hbar, \tilde{\beta})$ and $\mathcal{L}(K, \zeta, t)$ are $\hbar$-*level set* and $\zeta$-*level set* of $\left(K, \mathcal{C}_{(\hbar, \zeta)}\right)$ with level indices $\tilde{\beta}$ and $t$, respectively.

**Theorem 9.** *If $\left(K, \mathcal{C}_{(\hbar, \zeta)}\right)$ is a cCC-ideal of $K$, then its nonempty $\hbar$-level set and $\zeta$-level set are commutative ideals of $K$ for all level indices.*

**Proof.** Let $\tilde{\beta}$ and $t$ be level indices of $\left(K, \mathcal{C}_{(\hbar, \zeta)}\right)$ such that $\mathcal{L}(K, \hbar, \tilde{\beta})$ and $\mathcal{L}(K, \zeta, t)$ are nonempty. Then there exists $x \in \mathcal{L}(K, \hbar, \tilde{\beta}) \cap \mathcal{L}(K, \zeta, t)$, and so $\hbar(0) \succeq \hbar(x) \succeq \tilde{\beta}$ and $\zeta(0) \leq \zeta(x) \leq t$. Thus, $0 \in \mathcal{L}(K, \hbar, \tilde{\beta}) \cap \mathcal{L}(K, \zeta, t)$. Let $x, y, z \in K$ be such that $(x \div y) \div z \in \mathcal{L}(K, \hbar, \tilde{\beta}) \cap \mathcal{L}(K, \zeta, t)$ and $z \in \mathcal{L}(K, \hbar, \tilde{\beta}) \cap \mathcal{L}(K, \zeta, t)$. Then, $\hbar((x \div y) \div z) \succeq \tilde{\beta}$, $\hbar(z) \succeq \tilde{\beta}$, $\zeta((x \div y) \div z) \leq t$ and $\zeta(z) \leq t$. It follows from Equation (18) that

$$\hbar(x \div (y \wedge x)) \succeq \mathrm{rmin}\{\hbar((x \div y) \div z), \hbar(z)\} \succeq \tilde{\beta}$$

and $\zeta(x \div (y \wedge x)) \leq \max\{\zeta((x \div y) \div z), \zeta(z)\} \leq t$. Hence, $x \div (y \wedge x) \in \mathcal{L}(K, \hbar, \tilde{\beta}) \cap \mathcal{L}(K, \zeta, t)$. Therefore, $\mathcal{L}(K, \hbar, \tilde{\beta})$ and $\mathcal{L}(K, \zeta, t)$ are commutative ideals of $K$. □

**Corollary 3.** *If $\left(K, \mathcal{C}_{(\hbar, \zeta)}\right)$ is a cCC-ideal of $K$, then its nonempty level set $\mathcal{L}\left(K, \mathcal{C}_{(\hbar, \zeta)}, \tilde{\beta}, t\right)$ is a commutative ideal of $K$ for all $\tilde{\beta} := [\beta^-, \beta^+] \in [[0, 1]]$ and $t \in [-1, 0]$.*

**Theorem 10.** *Let $\left(K, \mathcal{C}_{(\hbar,\zeta)}\right)$ be a crossing cubic set structure on K in which its nonempty $\hbar$-level set and $\zeta$-level set are commutative ideals of K for all level indices. Then, $\left(K, \mathcal{C}_{(\hbar,\zeta)}\right)$ is a cCC-ideal of K.*

**Proof.** Assume that $\mathcal{L}(K, \hbar, \tilde{\beta})$ and $\mathcal{L}(K, \zeta, t)$ are nonempty commutative ideals of $K$ for all $\tilde{\beta} := [\beta^-, \beta^+] \in [[0,1]]$ and $t \in [-1, 0]$. Suppose $\left(K, \mathcal{C}_{(\hbar,\zeta)}\right)$ is not a cCC-ideal of $K$. Then at least one of Equations (17) and (18) is not valid. Suppose that Equation (18) is not valid. Then, there exist $a, b, x, y \in K$ be such that $\hbar(a \div (b \wedge a)) \npreceq \mathrm{rmin}\{\hbar((a \div b) \div c), \hbar(c)\}$ or $\zeta(x \div (y \wedge x)) \nleq \max\{\zeta((x \div y) \div z), \zeta(z)\}$. If we take $\tilde{\beta} := [\beta^-, \beta^+] = \mathrm{rmin}\{\hbar((a \div b) \div c), \hbar(c)\}$, then $(a \div b) \div c \in \mathcal{L}(K, \hbar, \tilde{\beta})$ and $c \in \mathcal{L}(K, \hbar, \tilde{\beta})$, but $a \div (b \wedge a) \notin \mathcal{L}(K, \hbar, \tilde{\beta})$. In addition, if we take $t = \max\{\zeta((x \div y) \div z), \zeta(z)\}$, then $(x \div y) \div z \in \mathcal{L}(K, \zeta, t)$ and $z \in \mathcal{L}(K, \zeta, t)$, but $x \div (y \wedge x) \notin \mathcal{L}(K, \zeta, t)$. This is a contradiction, and so Equation (18) is valid. In the same way, if Equation (17) is not established, it leads to a contradiction. Therefore, $\left(K, \mathcal{C}_{(\hbar,\zeta)}\right)$ is a cCC-ideal of $K$. $\square$

**Theorem 11.** *Given a subset L of K, we define a crossing cubic set structure $\left(K, \mathcal{C}_{(\hbar,\zeta)}\right)$ on K as follows:*

$$\hbar : K \to [[0,1]], \ x \mapsto \begin{cases} \tilde{\beta} := [\beta^-, \beta^+] & \text{if } x \in L, \\ \tilde{0} := [0,0] & \text{otherwise,} \end{cases}$$

$$\zeta : K \to [-1, 0], \ x \mapsto \begin{cases} t & \text{if } x \in L, \\ 0 & \text{otherwise,} \end{cases}$$

*where $\beta^-, \beta^+ \in (0,1]$ with $\beta^- < \beta^+$ and $t \in [-1, 0)$. Then L is a commutative ideal of K if and only if $\left(K, \mathcal{C}_{(\hbar,\zeta)}\right)$ is a cCC-ideal of K.*

**Proof.** The definition of $\left(K, \mathcal{C}_{(\hbar,\zeta)}\right)$ derives $\mathcal{L}(K, \hbar, \tilde{\beta}) = L$, $\mathcal{L}(K, \hbar, \tilde{0}) = K$, $\mathcal{L}(K, \zeta, t) = L$ and $\mathcal{L}(K, \zeta, 0) = K$. Hence, the desired result can be obtained by Theorems 9 and 10. $\square$

Note that a CC-ideal might not be a cCC-ideal (see Example 4), but we can consider the extension property for a cCC-ideal.

**Lemma 2** ([16])**.** *Let A and B be ideals of K such that $A \subseteq B$. If A is a commutative ideal of K, then so is B.*

**Theorem 12.** *Let $\left(K, \mathcal{C}_{(\hbar,\zeta)}\right)$ and $\left(K, \mathcal{C}_{(£,\eta)}\right)$ be CC-ideals of K such that*

$$\hbar(0) = £(0), \ \zeta(0) = \eta(0), \tag{29}$$
$$(\forall x(\neq 0) \in K)(\hbar(x) \preceq £(x), \ \zeta(x) \geq \eta(x)). \tag{30}$$

*If $\left(K, \mathcal{C}_{(\hbar,\zeta)}\right)$ is a cCC-ideal of K, then so is $\left(K, \mathcal{C}_{(£,\eta)}\right)$.*

**Proof.** Assume that $\left(K, \mathcal{C}_{(\hbar,\zeta)}\right)$ is a cCC-ideal of $K$. Then, its nonempty $\hbar$-level set and $\zeta$-level set are commutative ideals of $K$ for all level indices by Theorem 9, that is, $\mathcal{L}(K, \hbar, \tilde{\beta})$ and $\mathcal{L}(K, \zeta, t)$ are commutative ideals of $K$ whenever they are nonempty for all $\tilde{\beta} := [\beta^-, \beta^+] \in [[0,1]]$ and $t \in [-1, 0]$. It is clear from Equations (29) and (30) that $\mathcal{L}(K, \hbar, \tilde{\beta}) \subseteq \mathcal{L}(K, £, \tilde{\beta})$ and $\mathcal{L}(K, \zeta, t) \subseteq \mathcal{L}(K, \eta, t)$. As $\left(K, \mathcal{C}_{(£,\eta)}\right)$ is a CC-ideals of $K$, we know that $\mathcal{L}(K, £, \tilde{\beta})$ and $\mathcal{L}(K, \eta, t)$ are ideals of $K$. It follows from Lemma 2 that $\mathcal{L}(K, £, \tilde{\beta})$ and $\mathcal{L}(K, \eta, t)$ are commutative ideals of $K$. Therefore, $\left(K, \mathcal{C}_{(£,\eta)}\right)$ is a cCC-ideal of $K$ by Theorem 10. $\square$

## 5. Translations of Commutative Crossing Cubic Ideals

Given a crossing cubic set structure $\left(K, \mathcal{C}_{(\hbar, \zeta)}\right)$ on a set $K$, we denote

$$\bot := -1 - \inf\{\zeta(x) \mid x \in K\} \text{ and } \top := 1 - \sup\{\hbar^+(x) \mid x \in K\}. \tag{31}$$

For every $k \in [-1, 0]$ with $k \geq \bot$ and $\tilde{\beta} = [\beta^-, \beta^+] \in [[0, 1]]$ with $\beta^+ \leq \top$, we define

$$
\begin{aligned}
\hbar_{\tilde{\beta}}^T &: K \to [[0, 1]], \ x \mapsto \hbar(x) + \tilde{\beta}, \\
\zeta_k^T &: K \to [-1, 0], \ x \mapsto \zeta(x) + k.
\end{aligned}
\tag{32}
$$

Then, $\left(K, \mathcal{C}_{(\hbar_{\tilde{\beta}}^T, \zeta_k^T)}\right)$ is a crossing cubic set structure on $K$, which is called the $(\tilde{\beta}, k)$-*translation* of $\left(K, \mathcal{C}_{(\hbar, \zeta)}\right)$.

**Theorem 13.** *If* $\left(K, \mathcal{C}_{(\hbar, \zeta)}\right)$ *is a cCC-ideal of K, then its* $(\tilde{\beta}, k)$-*translation* $\left(K, \mathcal{C}_{(\hbar_{\tilde{\beta}}^T, \zeta_k^T)}\right)$ *is also a cCC-ideal of K for all* $\tilde{\beta} = [\beta^-, \beta^+] \in [[0, 1]]$ *and* $k \in [-1, 0]$ *with* $\beta^+ \leq \top$ *and* $k \geq \bot$, *respectively.*

**Proof.** Let $k \in [-1, 0]$ with $k \geq \bot$ and $\tilde{\beta} = [\beta^-, \beta^+] \in [[0, 1]]$ with $\beta^+ \leq \top$. Assume that $\left(K, \mathcal{C}_{(\hbar, \zeta)}\right)$ is a cCC-ideal of $K$ and let $x, y, z \in K$. Then,

$$\hbar_{\tilde{\beta}}^T(0) = \hbar(0) + \tilde{\beta} \succeq \hbar(x) + \tilde{\beta} = \hbar_{\tilde{\beta}}^T(x),$$

$$\zeta_k^T(0) = \zeta(0) + k \leq \zeta(x) + k = \zeta_k^T(x),$$

$$
\begin{aligned}
\hbar_{\tilde{\beta}}^T(x \div (y \wedge x)) &= \hbar(x \div (y \wedge x)) + \tilde{\beta} \succeq \mathrm{rmin}\{\hbar((x \div y) \div z), \hbar(z)\} + \tilde{\beta} \\
&= \mathrm{rmin}\{\hbar((x \div y) \div z) + \tilde{\beta}, \hbar(z) + \tilde{\beta}\} \\
&= \mathrm{rmin}\{\hbar_{\tilde{\beta}}^T((x \div y) \div z), \hbar_{\tilde{\beta}}^T(z)\}
\end{aligned}
$$

and

$$
\begin{aligned}
\zeta_k^T(x \div (y \wedge x)) &= \zeta(x \div (y \wedge x)) + k \leq \max\{\zeta((x \div y) \div z), \zeta(z)\} + k \\
&= \max\{\zeta((x \div y) \div z) + k, \zeta(z) + k\} \\
&= \max\{\zeta_k^T((x \div y) \div z), \zeta_k^T(z)\}.
\end{aligned}
$$

Hence, $\left(K, \mathcal{C}_{(\hbar_{\tilde{\beta}}^T, \zeta_k^T)}\right)$ is a cCC-ideal of $K$. □

**Theorem 14.** *Let* $\left(K, \mathcal{C}_{(\hbar, \zeta)}\right)$ *be a crossing cubic set structure on K. If there exist* $k \in [-1, 0]$ *and* $\tilde{\beta} = [\beta^-, \beta^+] \in [[0, 1]]$ *with* $k \geq \bot$ *and* $\beta^+ \leq \top$, *respectively, such that the* $(\tilde{\beta}, k)$-*translation* $\left(K, \mathcal{C}_{(\hbar_{\tilde{\beta}}^T, \zeta_k^T)}\right)$ *of* $\left(K, \mathcal{C}_{(\hbar, \zeta)}\right)$ *is a cCC-ideal of K, then* $\left(K, \mathcal{C}_{(\hbar, \zeta)}\right)$ *is a cCC-ideal of K.*

**Proof.** Assume that the $(\tilde{\beta}, k)$-translation $\left(K, \mathcal{C}_{(\hbar_{\tilde{\beta}}^T, \zeta_k^T)}\right)$ of $\left(K, \mathcal{C}_{(\hbar, \zeta)}\right)$ is a cCC-ideal of $K$ for some $\tilde{\beta} = [\beta^-, \beta^+] \in [[0, 1]]$ and $k \in [-1, 0]$ with $\beta^+ \leq \top$ and $k \geq \bot$, respectively. Then,

$\hbar(0) + \tilde{\beta} = \hbar_{\tilde{\beta}}^T(0) \succeq \hbar_{\tilde{\beta}}^T(x) = \hbar(x) + \tilde{\beta}$ and $\zeta(0) + k = \zeta_k^T(0) \leq \zeta_k^T(x) = \zeta(x) + k$ for all $x \in K$. Hence, $\hbar(0) \succeq \hbar(x)$ and $\zeta(0) \leq \zeta(x)$ for all $x \in K$. For every $x, y, z \in K$, we have

$$\hbar(x \div (y \wedge x)) + \tilde{\beta} = \hbar_{\tilde{\beta}}^T(x \div (y \wedge x)) \succeq \mathrm{rmin}\{\hbar_{\tilde{\beta}}^T((x \div y) \div z), \hbar_{\tilde{\beta}}^T(z)\}$$
$$= \mathrm{rmin}\{\hbar((x \div y) \div z) + \tilde{\beta}, \hbar(z) + \tilde{\beta}\}$$
$$= \mathrm{rmin}\{\hbar((x \div y) \div z), \hbar(z)\} + \tilde{\beta}$$

and

$$\zeta(x \div (y \wedge x)) + k = \zeta_k^T(x \div (y \wedge x)) \leq \max\{\zeta_k^T((x \div y) \div z), \zeta_k^T(z)\}$$
$$= \max\{\zeta((x \div y) \div z) + k, \zeta(z) + k\}$$
$$= \max\{\zeta((x \div y) \div z), \zeta(z)\} + k$$

which imply that $\hbar(x \div (y \wedge x)) \succeq \mathrm{rmin}\{\hbar((x \div y) \div z), \hbar(z)\}$ and $\zeta(x \div (y \wedge x)) \leq \max\{\zeta((x \div y) \div z), \zeta(z)\}$. Consequently, $\left(K, \mathcal{C}_{(\hbar, \zeta)}\right)$ is a cCC-ideal of $K$. □

Let $\left(K, \mathcal{C}_{(\hbar, \zeta)}\right)$ be a crossing cubic set structure on $K$ and consider $k \in [-1, 0]$ and $\tilde{\beta} = [\beta^-, \beta^+] \in [[0, 1]]$ with $k \geq \bot$ and $\beta^+ \leq \top$, respectively. We take a set

$$\mathcal{C}_{(\hbar, \zeta)}(\tilde{\gamma}, r) := \{x \in K \mid \hbar(x) \succeq \tilde{\gamma} - \tilde{\beta}, \zeta(x) \leq r - k\} \tag{33}$$

where $\tilde{\gamma} \in [[0, 1]]$ and $r \in [-1, 0]$ such that $\tilde{\gamma} \succ \tilde{\beta}$, $\gamma^+ \leq \top$ and $r < k$.

**Theorem 15.** *If $\left(K, \mathcal{C}_{(\hbar, \zeta)}\right)$ is a cCC-ideal of $K$, then the set $\mathcal{C}_{(\hbar, \zeta)}(\tilde{\gamma}, r)$ in (33) is a commutative ideal of $K$ for all $\tilde{\gamma} \in [[0, 1]]$ and $r \in [-1, 0]$ such that $\tilde{\gamma} \succ \tilde{\beta}$, $\gamma^+ \leq \top$ and $r < k$.*

**Proof.** Suppose that $\left(K, \mathcal{C}_{(\hbar, \zeta)}\right)$ is a cCC-ideal of $K$ and let $x, y, z \in K$ be such that $(x \div y) \div z \in \mathcal{C}_{(\hbar, \zeta)}(\tilde{\gamma}, r)$ and $z \in \mathcal{C}_{(\hbar, \zeta)}(\tilde{\gamma}, r)$. Then, $\hbar((x \div y) \div z) \succeq \tilde{\gamma} - \tilde{\beta}$, $\zeta((x \div y) \div z) \leq r - k$, $\hbar(z) \succeq \tilde{\gamma} - \tilde{\beta}$, and $\zeta(z) \leq r - k$. It follows from Equation (18) that $\hbar(x \div (y \wedge x)) \succeq \mathrm{rmin}\{\hbar((x \div y) \div z), \hbar(z)\} \succeq \tilde{\gamma} - \tilde{\beta}$ and $\zeta(x \div (y \wedge x)) \leq \max\{\zeta((x \div y) \div z), \zeta(z)\} \leq r - k$. Thus, $x \div (y \wedge x) \in \mathcal{C}_{(\hbar, \zeta)}(\tilde{\gamma}, r)$. It is clear that $0 \in \mathcal{C}_{(\hbar, \zeta)}(\tilde{\gamma}, r)$. Therefore, $\mathcal{C}_{(\hbar, \zeta)}(\tilde{\gamma}, r)$ is a commutative ideal of $K$. □

We explore the conditions under which the $(\tilde{\beta}, k)$-translation of a crossing cubic set structure on $K$ becomes a cCC-ideal of $K$.

**Theorem 16.** *Let $\left(K, \mathcal{C}_{(\hbar, \zeta)}\right)$ be a crossing cubic set structure on $K$. Then, the $(\tilde{\beta}, k)$-translation $\left(K, \mathcal{C}_{(\hbar_{\tilde{\beta}}^T, \zeta_k^T)}\right)$ of $\left(K, \mathcal{C}_{(\hbar, \zeta)}\right)$ is a cCC-ideal of $K$ if, and only if, the nonempty sets*

$$\mathcal{C}_{(\hbar, \zeta)}(\tilde{\gamma}) := \{x \in K \mid \hbar(x) \succeq \tilde{\gamma} - \tilde{\beta}\} \text{ and } \mathcal{C}_{(\hbar, \zeta)}(r) := \{x \in K \mid \zeta(x) \leq r - k\}$$

*are commutative ideals of $K$ for all $\tilde{\gamma} \in [[0, 1]]$ and $r \in [-1, 0]$ such that $\tilde{\gamma} \succ \tilde{\beta}$, $\gamma^+ \leq \top$ and $r < k$.*

**Proof.** Assume that $\left(K, \mathcal{C}_{(\hbar_{\tilde{\beta}}^T, \zeta_k^T)}\right)$ is a cCC-ideal of $K$. Let $x \in \mathcal{C}_{(\hbar, \zeta)}(\tilde{\gamma})$ and $a \in \mathcal{C}_{(\hbar, \zeta)}(r)$ for all $\tilde{\gamma} \in [[0, 1]]$ and $r \in [-1, 0]$ such that $\tilde{\gamma} \succ \tilde{\beta}$, $\gamma^+ \leq \top$ and $r < k$. Then,

$$\hbar(0) + \tilde{\beta} = \hbar_{\tilde{\beta}}^T(0) \succeq \hbar_{\tilde{\beta}}^T(x) = \hbar(x) + \tilde{\beta} \succeq \tilde{\gamma} - \tilde{\beta} + \tilde{\beta} = \tilde{\gamma}$$

and

$$\zeta(0) + k = \zeta_k^T(0) \leq \zeta_k^T(a) = \zeta(a) + k \leq r - k + k = r,$$

which shows that $0 \in \mathcal{C}_{(\hbar,\zeta)}(\tilde{\gamma}) \cap \mathcal{C}_{(\hbar,\zeta)}(r)$. Let $x, y, z, a, b, c \in K$ be such that $(x \div y) \div z \in \mathcal{C}_{(\hbar,\zeta)}(\tilde{\gamma})$, $z \in \mathcal{C}_{(\hbar,\zeta)}(\tilde{\gamma})$, $(a \div b) \div c \in \mathcal{C}_{(\hbar,\zeta)}(r)$ and $c \in \mathcal{C}_{(\hbar,\zeta)}(r)$. Then,

$$\begin{aligned}
\hbar(x \div (y \wedge x)) + \tilde{\beta} = \hbar_{\tilde{\beta}}^T(x) &\succeq \mathrm{rmin}\{\hbar_{\tilde{\beta}}^T((x \div y) \div z), \hbar_{\tilde{\beta}}^T(z)\} \\
&= \mathrm{rmin}\{\hbar((x \div y) \div z) + \tilde{\beta}, \hbar(z) + \tilde{\beta}\} \\
&= \mathrm{rmin}\{\hbar((x \div y) \div z), \hbar(z)\} + \tilde{\beta} \\
&\succeq \tilde{\gamma} - \tilde{\beta} + \tilde{\beta} = \tilde{\gamma}
\end{aligned}$$

and

$$\begin{aligned}
\zeta(a \div (b \wedge a)) + k = \zeta_k^T(a \div (b \wedge a)) &\leq \max\{\zeta_k^T((a \div b) \div c), \zeta_k^T(c)\} \\
&= \max\{\zeta((a \div b) \div c) + k, \zeta(c) + k\} \\
&= \max\{\zeta((a \div b) \div c), \zeta(c)\} + k \\
&\leq r - k + k = r.
\end{aligned}$$

Hence, $x \div (y \wedge x) \in \mathcal{C}_{(\hbar,\zeta)}(\tilde{\gamma})$ and $a \div (b \wedge a) \in \mathcal{C}_{(\hbar,\zeta)}(r)$. Therefore, $\mathcal{C}_{(\hbar,\zeta)}(\tilde{\gamma})$ and $\mathcal{C}_{(\hbar,\zeta)}(r)$ are commutative ideals of $K$.

Conversely, suppose that $\mathcal{C}_{(\hbar,\zeta)}(\tilde{\gamma})$ and $\mathcal{C}_{(\hbar,\zeta)}(r)$ are commutative ideals of $K$ for all $\tilde{\gamma} \in [[0,1]]$ and $r \in [-1,0]$ such that $\tilde{\gamma} \succ \tilde{\beta}$, $\gamma^+ \leq \top$ and $r < k$. If there exists $(x, a) \in K \times K$ such that $\hbar_{\tilde{\beta}}^T(0) \succeq \hbar_{\tilde{\beta}}^T(x)$ or $\zeta_k^T(0) \leq \zeta_k^T(a)$ are not true, then $\hbar_{\tilde{\beta}}^T(0) \prec \tilde{\delta} \preceq \hbar_{\tilde{\beta}}^T(x)$ or $\zeta_k^T(0) > s \geq \zeta_k^T(a)$ for some $\tilde{\delta} \in [[0,1]]$ and $s \in [-1,0]$ such that $\tilde{\delta} \succ \tilde{\beta}$, $\delta^+ \leq \top$ and $s < k$. Then, $0 \notin \mathcal{C}_{(\hbar,\zeta)}(\tilde{\gamma})$ or $0 \notin \mathcal{C}_{(\hbar,\zeta)}(r)$, which is a contradiction and so $\hbar_{\tilde{\beta}}^T(0) \succeq \hbar_{\tilde{\beta}}^T(x)$ and $\zeta_k^T(0) \leq \zeta_k^T(x)$ for all $x \in K$. If there exist $(a, b, c), (x, y, z) \in K \times K \times K$ such that $\hbar_{\tilde{\beta}}^T(a \div (b \wedge a)) \succeq \mathrm{rmin}\{\hbar_{\tilde{\beta}}^T((a \div b) \div c), \hbar_{\tilde{\beta}}^T(c)\}$ or $\zeta_k^T(x \div (y \wedge x)) \leq \max\{\zeta_k^T((x \div y) \div z), \zeta_k^T(z)\}$ are not true, then $\hbar_{\tilde{\beta}}^T(a \div (b \wedge a)) \prec \tilde{\delta} \preceq \mathrm{rmin}\{\hbar_{\tilde{\beta}}^T((a \div b) \div c), \hbar_{\tilde{\beta}}^T(c)\}$ or $\zeta_k^T(x \div (y \wedge x)) > s \geq \max\{\zeta_k^T((x \div y) \div z), \zeta_k^T(z)\}$ for some $\tilde{\delta} \in [[0,1]]$ and $s \in [-1,0]$ such that $\tilde{\delta} \succ \tilde{\beta}$, $\delta^+ \leq \top$ and $s < k$. It follows that

$$\hbar((a \div b) \div c) \succeq \tilde{\delta} - \tilde{\beta}, \ \hbar(c) \succeq \tilde{\delta} - \tilde{\beta}$$

or

$$\zeta((x \div y) \div z) \leq s - k, \ \zeta(z) \leq s - k,$$

that is, $(a \div b) \div c, c \in \mathcal{C}_{(\hbar,\zeta)}(\tilde{\gamma})$ or $(x \div y) \div z, z \in \mathcal{C}_{(\hbar,\zeta)}(r)$. However, $\hbar(a \div (b \wedge a)) \not\succeq \tilde{\delta} - \tilde{\beta}$ or $\zeta(x \div (y \wedge x)) \not\leq s - k$, that is, $a \div (b \wedge a) \notin \mathcal{C}_{(\hbar,\zeta)}(\tilde{\gamma})$ or $x \div (y \wedge x) \notin \mathcal{C}_{(\hbar,\zeta)}(r)$. This is a contradiction, and thus $\hbar_{\tilde{\beta}}^T(a \div (b \wedge a)) \succeq \mathrm{rmin}\{\hbar_{\tilde{\beta}}^T((a \div b) \div c), \hbar_{\tilde{\beta}}^T(c)\}$ and $\zeta_k^T(x \div (y \wedge x)) \leq \max\{\zeta_k^T((x \div y) \div z), \zeta_k^T(z)\}$ for all $a, b, c, x, y, z \in K$. Therefore, $\left(K, \mathcal{C}_{(\hbar_{\tilde{\beta}}^T, \zeta_k^T)}\right)$ is a cCC-ideal of $K$. □

## 6. Conclusions and Future Work

In this manuscript, we provided two binary operations to assign a semigroup structure to the set of crossing cubic set structures and applied the crossing cubic set structure to commutative ideals of BCK-algebras and defined commutative crossing cubic ideal. We investigated several properties and the relationship between crossing cubic ideal and commutative crossing cubic ideal. We presented an example in which crossing cubic ideal is not commutative crossing cubic ideal, and then we explored the conditions in which crossing

cubic ideal can be commutative crossing cubic ideal. We discussed characterizations of commutative crossing cubic ideal and investigated the relationship between commutative crossing cubic ideal and crossing cubic level set. We established an extension property of commutative crossing cubic ideal and studied the translation of commutative crossing cubic ideal. We found conditions for the translation of crossing cubic set structure to be commutative crossing cubic ideal and considered its characterization.

Our future work involves applications of crossing cubic set structure to substructures of various algebraic structures, for example, hoop algebra, equality algebra, EQ-algebra, BL-algebra, BE-algebra, group, (near, semi) ring, etc. Based on these studies, we will find ways and technologies to apply K to decision-making theory, computer science, medical science, etc., in the future.

**Author Contributions:** Create and conceptualize ideas: Y.B.J.; writing—original draft preparation, Y.B.J. and M.A.Ö.; writing—review and editing, M.A.Ö. and D.Y. All authors have read and agreed to the published version of the manuscript.

**Funding:** This research received no external funding.

**Acknowledgments:** The authors wish to thank the anonymous reviewers for their valuable suggestions.

**Conflicts of Interest:** The authors declare no conflict of interest.

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
