# Peer review of "Semigroup Structures and Commutative Ideals of BCK-Algebras Based on Crossing Cubic Set Structures"

_axioms, doi:10.3390/axioms11010025_

Round 1
Reviewer 1 Report
Please see the attached pdf. file

Author Response
Revision Note
Manuscript ID: axioms-1497786
Title: Semigroup structures and commutative ideals of BCK-algebras based on crossing cubic set structures
Authors: Mehmet Ali ÖZTÜRK, Damla Yilmaz , Young Bae Jun
Reviewer 1 (Round 1):
The paper is scientifically correct and its level is good for people working in related fields. The bibliography is reasonable. The abstract gives a correct description of the content. Also, with respect to the content, the length of the paper is reasonable.
There are some minor observations which should be taken into consideration:
- Pg. 11 line 124, instead of \Then at least on of\ should be written \Then at least one of\
- Pg. 12 line 134, instead of word \desied\ should be written \desired\
- Pg. 12 line 135, after “see Example 4.” is missing a bracket “)”
- Pg. 15 line 175. After the words “In this manuscript,” it follows the word We, which should be with lowercase we. Also in the Conclusions section it should removed the comma before the word ”, and”
Answer: We revised everything you pointed out. Thank you for your careful reading.
To conclude: The authors should revise the paper and make the appropriate corrections. After that, I recommend to publish this paper in Axioms (ISSN 2075-1680).
Answer: We appreciate your delicate point of view.
Thank you very much for your meticulous reading and your comments to make it a more up-to-date paper.
14 December 2021
- A. ÖZTÜRK (on behalf of co-authors)

Reviewer 2 Report
The authors propose a concept of commutative crossing cubic ideal by applying crossing cubic set structure to commutative ideal in BCK-algebra, and several properties are investigated. Then, the authors discussed the relationship between crossing cubic ideal and commutative crossing cubic ideal. The paper is well written. However, before accepting some shortcomings must be eliminated. The list of comments is as follows:
1. The literature review is not enough.
2. The introduction should be extended to better show the novelty and contribution of this paper.
3. the authors make very short descriptions in the rest of the papers. Some proofs can be put to the appendix to the more clear narration. However, please extend the description.
4. te language should be improved also.
Author Response
Revision Note
Manuscript ID: axioms-1497786
Title: Semigroup structures and commutative ideals of BCK-algebras based on crossing cubic set structures
Authors: Mehmet Ali ÖZTÜRK, Damla Yilmaz , Young Bae Jun
Reviewer 2 (Round 1):
The authors propose a concept of commutative crossing cubic ideal by applying crossing cubic set structure to commutative ideal in BCK-algebra, and several properties are investigated. Then, the authors discussed the relationship between crossing cubic ideal and commutative crossing cubic ideal. The paper is well written. However, before accepting some shortcomings must be eliminated. The list of comments is as follows:
1. The literature review is not enough.
- The introduction should be extended to better show the novelty and contribution of this paper.
Answer: We did our best to write the literature review and introduction required for this paper. However, we reviewed it again because you said you were lacking, but we couldn't decide in detail what part to add. You are a professional reviewer who reviewed this paper. So we would appreciate it if you could tell us in detail what you feel is lacking after reviewing it. If you think readers have no inconvenience reading this paper as it is, we would appreciate it if you could understand it deeply. If you think it is difficult to read this paper due to a lack of literature review and introduction, we would appreciate it if you could let us know in detail what is lacking.
- the authors make very short descriptions in the rest of the papers. Some proofs can be put to the appendix to the more clear narration. However, please extend the description.
Answer: We are very sorry we didn't fully understand your comment. We think your suggestion is about the composition or narrative method of the text description, but we wrote the paper according to the universal narrative method of the (applied) mathematics papers. You are the expert who reviewed this paper, so we think we should respect all your suggestions and follow them as much as possible. That is because your comment plays a decisive role in improving the quality of this paper. However, if the reviewer's proposal is not specific, it is not easy for the authors to know the reviewer's intentions. We would appreciate it if you could let us know in detail what you are asking for in this proposal. If possible, we would appreciate it if you could send us a sample of the (applied) mathematics paper that was written in accordance with your request. Then, with gratitude, we will be able to modify the paper in the form you want.
If you look at this paper again and think it is a general form as it is in (applied) mathematics papers, we would appreciate it if you could recommend it as it is and encourage the authors.
- the language should be improved also.
Answer: We tried to speak the language as grammatically as possible, but we reviewed the entire paper again according to your advice.
Since we cannot say that English expressions are unique, we think there will still be some parts that you may feel insufficient from your point of view.
Even if it's not your preferred expression, we would appreciate it if you could understand it generously if it wasn't grammatically wrong and if there was no fear of distortion of the paper.
However, if there is something you don't understand and it doesn't pass or there is a grammatical error, please let us know in detail and we will respect your opinion and revise it.
Thank you very much for your meticulous reading and your comments to make it a more up-to-date paper.
14 December 2021
- A. ÖZTÜRK (on behalf of co-authors)

Round 2
Reviewer 2 Report
I encourage the authors to review Axioms' published articles to understand this comment better. Currently, the article more closely resembles a conference draft. 14 References I also consider not very applicable. Please improve the paper according to my previous suggestions.
Author Response
Revision Note
Manuscript ID: axioms-1497786
Title: Semigroup structures and commutative ideals of BCK-algebras based on crossing cubic set structures
Authors: Mehmet Ali ÖZTÜRK, Damla Yilmaz , Young Bae Jun
Reviewer 2 (Round 2): .
I encourage the authors to review Axioms' published articles to understand this comment better. Currently, the article more closely resembles a conference draft. 14 References I also consider not very applicable. Please improve the paper according to my previous suggestions..
Answer: We supplemented the introduction and references. The supplemented part is highlighted in yellow in the text.
We looked closely at Axioms' published articles according to your suggestion. Since this paper is based on pure mathematics, especially algebra, we have focused on it, but we have not found anything special, and we think our paper is also in the category of Axioms' published papers.
01 January 2022
M. A. ÖZTÜRK (on behalf of co-authors)

Round 3
Reviewer 2 Report
The paper has been improved and can be accepted in its current form.